# The Role of Nucleocytoplasmic Transport Defects in Amyotrophic Lateral Sclerosis

**DOI:** 10.3390/ijms222212175

**Published:** 2021-11-10

**Authors:** Joni Vanneste, Ludo Van Den Bosch

**Affiliations:** 1Experimental Neurology, Department of Neurosciences and Leuven Brain Institute (LBI), KU Leuven–University of Leuven, B-3000 Leuven, Belgium; Joni.Vanneste@kuleuven.be; 2Laboratory of Neurobiology, Center for Brain & Disease Research, VIB, B-3000 Leuven, Belgium

**Keywords:** neurodegeneration, motor neuron, ALS

## Abstract

There is ample evidence that nucleocytoplasmic-transport deficits could play an important role in the pathology of amyotrophic lateral sclerosis (ALS). However, the currently available data are often circumstantial and do not fully clarify the exact causal and temporal role of nucleocytoplasmic transport deficits in ALS patients. Gaining this knowledge will be of great significance in order to be able to target therapeutically nucleocytoplasmic transport and/or the proteins involved in this process. The availability of good model systems to study the nucleocytoplasmic transport process in detail will be especially crucial in investigating the effect of different mutations, as well as of other forms of stress. In this review, we discuss the evidence for the involvement of nucleocytoplasmic transport defects in ALS and the methods used to obtain these data. In addition, we provide an overview of the therapeutic strategies which could potentially counteract these defects.

## 1. Introduction

In eukaryotic cells, genomic material is separated from other intracellular compartments by the nuclear envelope. This compartmentalization results in a physical separation between transcription and translation processes, which allows the cell to strictly regulate gene expression [1]. This physical separation requires selective transport along the nuclear membrane. For instance, mRNAs and cytoplasmic proteins leaked into the nucleus need to be exported into the cytoplasm. On the contrary, proteins important for nuclear functions need to undergo nuclear import. The nucleocytoplasmic transport machinery consists of three major components: nuclear pore complexes, nuclear transport receptors and a RanGTP gradient. Mobile nuclear transport receptors will bind their cargo in a Ran-dependent way and facilitate their transport across the nuclear pore complex. During the last few years, substantial evidence was presented supporting the involvement of impaired nucleocytoplasmic transport in amyotrophic lateral sclerosis (ALS).

ALS is a neurodegenerative disorder characterized by the selective death of motor neurons in the motor cortex, the brainstem and the ventral horn of the spinal cord [2]. This leads to progressive paralysis and death of the patient on average 2 to 5 years after the detection of the first symptoms. In 90% of cases, ALS is a sporadic disease (sporadic ALS; sALS) while in 10% of patients there is a clear genetic cause of the disease which is dominantly inherited in almost all cases (familial ALS; fALS). The most important genetic causes of ALS are mutations in the genes encoding superoxide 1 (SOD1), TAR DNA binding protein 43 (TDP-43), fused in sarcoma (FUS) and hexanucleotide repeats in a non-coding region of the *C9orf72* gene.

Treatment options for ALS remain essentially supportive, since no effective cure is available. The fact that the only drugs on the market, riluzole (Rilutek) and edavarone (Radicava), provide only modest benefits and only in some patients [3,4], emphasizes the need for the development of new therapeutic strategies.

## 2. Genetic Causes of ALS

Despite the fact that rare mutations are found in many different genes (e.g., *VAPB, UBQLN2*, *OPTN, KIF5A*, …), almost all the information documenting a potential role of nucleocytoplasmic transport defects in ALS was obtained using models related to the four major genetic cases of fALS. Therefore, we will shortly introduce each of these major genetic causes.

### 2.1. Superoxide Dismutase 1

*SOD1* mutations were the first identified genetic cause of ALS, discovered in 1993 [5]. SOD1 is a cytoplasmic enzyme that converts reactive superoxide, which is produced during cellular respiration, into hydrogen peroxide [6]. More than 150 mutations dispersed all over the gene have been reported [7]. Almost all are dominant missense mutations and the disease is caused by a gain of function.

### 2.2. TAR DNA Binding Protein 43

TDP-43 is a DNA/RNA binding protein and is mainly concentrated in the nucleus, where it regulates transcription and splicing. A small fraction of TDP-43 is found in the cytoplasm where the protein plays a role in mRNA stability, transport and translation (for reviews: [8,9]). The majority of mutations are missense mutations that reside in the glycine-rich C-terminal region. This region contains an aggregation prone prion-like domain and disease-associated variations are thought to induce increased aggregation propensities. In addition, fragmented forms of TDP-43 containing the C-terminal region are prevalent in cytoplasmic inclusions seen in ALS pathology (for a review: [10]).

### 2.3. Fused in Sarcoma

Similar to TDP-43, FUS is mainly localized in the nucleus and can also shuttle to and from the cytoplasm [11], and plays a role in transcription, RNA splicing, RNA transport and translation (for reviews: [8,9]). Mutations in FUS are predominantly present in the N-terminal glycine-rich regions and in the C-terminal nuclear localization signal (NLS) [12]. There is a synergistic effect between the N-terminal prion-like domain and the C-terminal arginine-rich domain mediating FUS toxicity [13].

### 2.4. Chromosome 9 Open Reading Frame 72

More recently, GGGGCC hexanucleotide repeats (G_4_C_2_) were found in *C9orf72* [14,15,16]. This is the most common genetic cause of ALS in Europe and North America [17]. Moreover, this is also an important genetic cause of familial frontotemporal dementia (FTD), a neurodegenerative disease belonging to the same disease spectrum as ALS [18,19].

The *C9orf72* gene produces two protein isoforms, a short (C9orf72-S, 24 kDa) and a long isoform (C9orf72-L, 54 kDa). Based on bioinformatic analysis, a structural similarity between full-length C9orf72 and ‘Differentially Expressed in Normal and Neoplastic cells’ (DENN)-like proteins was shown [20,21]. DENN-domain containing proteins function as GDP/GTP exchange factors (GEFs) which activate Rab-GTPase-mediated membrane trafficking. Therefore, C9orf72-L could function as a GEF involved in regulating endolysosomal trafficking and autophagy [22,23,24,25].

A different biological function has been suggested for the short variant of the C9orf72 protein, as a different subcellular localization was observed [26]. C9orf72-S was found to be mainly localized at the nuclear membrane, suggesting its involvement in nucleocytoplasmic transport [26]. However, the relevance of this finding is debated because another study, which made use of knock-out validated antibodies, did not detect the C9orf72-S isoform in the human central nervous system (CNS) [27], indicating that C9orf72-L is the predominantly expressed isoform.

Three disease mechanisms have been proposed to play a role in mutant *C9orf72* induced neurodegeneration. Initially, it was proposed that the presence of the repeat expansion suppressed the production of the C9orf72 protein, although it remains to be proven whether this plays indeed an important role in the disease process (for a review: [28]). In contrast to this loss of function mechanism, two toxic gain of function mechanisms have been proposed. First, the repeat RNA itself could cause direct toxicity by sequestering, and consequently depleting essential RNA binding proteins in nuclear RNA foci (for a review: [29]). This could disturb RNA processing [30] and translation [31] and induce nucleolar stress [32]. Second, toxicity could arise from dipeptide repeat proteins (DPRs) that are translated from the repeat RNA by RAN translation. The exact pathological mechanisms by which DPRs could be toxic is a topic of intensive investigations (for a review: [33]). Multiple studies observed quite potent toxicity induced by poly-GR and poly-PR expression in *Drosophila* models, while little or no toxicity was caused by the other DPRs. Mainly for this reason, poly-GR and poly-PR are generally considered as the most toxic DPRs [34,35,36,37]. The strong toxic potential of poly-GR and poly-PR could arise from their highly charged and polar nature. Consequently, they could interact with many different proteins and were suggested to disturb multiple cellular functions [0]. Indeed, a wide range of proteins have been shown to interact with poly-GR/PR peptides in several proteomics experiments [38,39,40,41,42,43,44]. In the context of nucleocytoplasmic transport, interactions with both FG-nups and importins were observed [41,45,46,47].

## 3. ALS Histopathology

Despite the heterogeneity in ALS, TDP-43 is mislocalized to the cytoplasm in the majority of ALS patients [48]. Indeed, TDP-43 is depleted from the nucleus and forms cytoplasmic aggregates in ~97% of ALS patients [48]. This is intriguing as only a low number of patients carry a mutation in the gene encoding TDP-43 (*TARDBP*). For this reason, TDP-43 has become notorious in the ALS field as being the main culprit of the diseases. These pathological forms of TDP-43 exist of ubiquitinated, hyperphosphorylated and cleaved forms of the protein [48]. This pathology is evident in both neurons and glial cells from various brain regions [49].

TDP-43 pathology is absent in patients carrying a disease-causing *FUS* or *SOD1* mutation [14,50,51,52], although there is some controversy about this issue [53,54]. In the patients with disease-causing *FUS* or *SOD1* mutations, the corresponding proteins aggregate. Analogous to mutant TDP-43, ALS patients with pathogenic variants in FUS are characterized by cytoplasmic aggregation of FUS [50,55]. Moreover, the degree of cytoplasmic mislocalization of FUS correlates with the severity of the clinical presentation and age of onset [56,57,58].

## 4. Nucleocytoplasmic Transport

### 4.1. The Nuclear Pore Complex (NPC)

#### 4.1.1. Structure

Nuclear pore complexes are large multi-protein structures that perforate the nuclear membrane. They have a molecular mass of 110 MDa in humans and have an eightfold radial symmetry [59,60]. These pores are grossly symmetric along the plane of the membrane, but the peripheral components on the nuclear and cytoplasmic faces are different (Figure 1) [59].

Nucleoporins (nups) are the building block of NPCs. There are ~30 different nups and, owing to the eightfold symmetry of the pores, each nucleoporin is present in copies of eight or multiples of eight. This results in 500 to 1000 nups per pore [59].

Nups can be classified into distinct subcomplexes (Figure 1) [60,61,62]. First, transmembrane nucleoporins reside in the nuclear membrane and anchor the nuclear pore to the membrane [60,62]. Three transmembrane nucleoporins are known in humans: ndc1, pom121 and nup210 [60]. Second, scaffold nucleoporins comprise the core backbone of the NPC and exist of mainly two groups: a nup107–nup160 complex (or Y complex) forming the outer ring and a nup93–nup205 complex forming the inner ring [61]. The inner ring is sandwiched between two outer rings, one on the nucleocytoplasmic and one on the nuclear side [61]. This triple-ring framework creates a central transport channel with a diameter of about 35 nm [61]. Third, nucleoporins that line the surface of the central tube, all the way from the nucleus to the cytoplasm, harbor characteristic phenylalanine (FG)-rich repeats. This group exists of nucleoporins of the cytoplasmic filaments (e.g., nup214 and nup358), the nuclear basket (e.g., nup153) and the central channel (e.g., nup62 and nup98) [62]. The main function of these nups is mediating the selectivity and permeability of the NPC [64]. Fourth, linker nups create a bridge between the core scaffold and the central FG nups (e.g., nup93) [61].

#### 4.1.2. Functions

NPCs are gigantic molecular machines that operate as the main transportation hub between the nucleus and cytoplasm. Importantly, these pores are highly selective and, as such, they maintain the distinct molecular composition of the nucleus and the cytoplasm [64].

The basis of this selective permeability barrier are the FG domains of the FG nups that project into the central channel of the pore (Figure 1). These FG domains are repetitive and often contain up to 50 FG motifs [64]. Moreover, they have a low sequence complexity and are intrinsically disordered [64]. Uniquely, these flexible domains are also depleted from charged residues [64]. As a consequence, different FG nups will crosslink with each other due to multivalent, hydrophobic and low-affinity interactions. This results in the formation of a dense polymer meshwork [64]. This sieve-like structure is freely permeable for objects smaller then ~30 kDa or ~5 nm, but becomes increasingly restrictive for larger molecules [64]. Transport of these molecules is facilitated with the help of NTRs, as explained below.

Peripheral nups also play a role in ensuring a smooth flow in and out the nucleus. For example, cytoplasmic filaments anchor the NPC to the microtubule cytoskeleton, which may help to keep cargoes ‘on track’ as they cross the nuclear membrane (Figure 1) [61]. In addition, cytoplasmic filaments link the NPC with the protein synthesis machinery, facilitating a close coupling between export of messenger ribonucleic proteins (mRNPs) and translation initiation (Figure 1) [61]. On the nuclear side, the nuclear basket seems to be involved in preventing defective mRNA, e.g., unspliced mRNAs, from leaving the nucleus [61]. In addition, a filamentous lattice is formed that interlinks the nuclear baskets of neighboring NPCs [61], also known as the nuclear lamina. It has been suggested that this platform might play a role in excluding large macromolecular assemblies, including heterochromatin, from the vicinity of the NPC entrance. This would ensure a blockage-free nuclear transport [61].

### 4.2. Nuclear Transport Receptors (NTRs)

#### 4.2.1. Introduction

NTRs are soluble receptors that mediate transport of their cargoes between the nucleus and cytoplasm, so called facilitated or active transport [65]. They bind cargoes on one side of the nuclear membrane, cross the NPC barrier, release the cargoes on the other side and return for the next round. As explained before, a dense network of intertwined FG repeat filaments is present in the central channel of the NPC. This forms an obstacle for passive diffusion of most molecules. Importantly, NTR have the unique ability to overcome this permeability barrier through their capacity to interact with FG repeats (Figure 1) [64]. By binding to the FG domains, they transiently and locally open individual meshes of the network to facilitate passage of bound cargoes. These interactions are very weak, multivalent and fast [64]. As a consequence, NPC passage only takes a few milliseconds [66]. This allows NPCs to have a tremendous transport capacity of nearly 1000 NTR molecules per pore per second [67].

#### 4.2.2. Functions

The majority of NTRs belong to the family of karyopherin β family (KPNB), which has at least 20 members in humans [65]. β-karyopherins that mediate nuclear export are known as exportins and those mediating nuclear import are known as importins.

Exportins bind the nuclear export signal (NES) of their cargoes localized in the nucleus. They subsequently move through the central channel of the NPC and release their cargo in the cytoplasm before returning to the nucleus (Figure 2A).

Several pathways of nuclear export have been identified, but the most common type of export is mediated by exportin 1 (XPO1, also known as Crm1). Exportin 1 recognizes a leucine-rich NES [65], of which the NES from protein kinase inhibitor (PKI) is the most often used prototype [68]. More than 1000 different proteins are thought to be exported by exportin 1 in human cells [69]. This high number of cargoes indicates a very broad impact of this nuclear export pathway on cellular physiology [69]. In addition, most of these proteins have a very strong bias towards a cytoplasmic localization [69]. This suggests that exportin 1 plays an important role in back-sorting leaked cytoplasmic proteins from the nucleus [69].

Nuclear import is mediated by importins that bind proteins containing a nuclear localization signal (NLS). Various receptor-mediated import pathways have been identified, but the best characterized pathway involves karyopherin-β1 (KPNB1, also known as importin-β1). Importin-β1 binds its cargo directly or via adapter molecules [70,71]. The best studied adapter is importin-α (also known as karyopherin-α) which binds importin-β1 via its N-terminal importin β-binding domain (IBB) [65]. Importin-α binds cargoes containing a classical NLS (cNLS) [70,71].

Classical NLSs exist of either one (monopartite, e.g., NLS_SV40_ PKKKRRV) or two (bipartite, e.g., NLS_nucleoplasmin_ KR-PAATKKAGQA-KKKK) stretches of basic amino acids [72]. As a consequence, these NLSs are highly concentrated with arginines (R) and lysines (K). Of interest, the importin-β1/importin-α1 (KPNB1/KPNA2) pathway is used by TDP-43, the protein mislocalized in the majority of ALS patients [48].

Import via karyopherin β2 (also known as transportin 1 or importin-β2) is another major import pathway (Figure 2B) [73]. Transportin 1 is responsible for the import of the ALS-protein FUS [56]. These proteins contain a PY-NLS, named after the R/K/Hx_2–5_PY motif found towards the C-terminus of the NLS [74]. Similarly to the cNLS, the arginine and lysine residues are structurally conserved and are important for transportin binding [75].

Besides their import function, importins also have an important task as chaperones. Importins were observed to effectively suppress the aggregation of their cargoes in a polyanionic environment (e.g., presence of tRNA) [76]. They achieved this by shielding the exposed basic residues of their cargoes [76]. As this chaperoning function needs a precise match between cargo and receptor, this could explain the high number of existing importins [76].

Recent work of multiple groups demonstrated the importance of this chaperoning function to protect against pathological protein aggregation in ALS, as both transportin 1 (TPNO1) and importin-α/β prevented aggregation of FUS and TDP-43, respectively [77,78,79,80]. Furthermore, these importins have the ability to shield the toxic arginine-rich DPRs, poly-PR and poly-GR. As such, it is suggested that importins have the ability to buffer the pathological interactions of these peptides. For example, these importins suppressed the pathological interaction of poly-PR and poly-GR with TDP-43 and decreased poly-PR and poly-GR induced insolubility of TDP-43.

### 4.3. The RanGTP Gradient

#### 4.3.1. Introduction

The Ran protein is a Ras-related GTPase that can switch between a GTP-bound state (RanGTP) and a GDP-bound state (RanGDP) [81]. Importantly, these two different nucleotide-bound forms are present in the cell in a gradient. RanGTP is present at high concentrations in the nucleus, while the cytoplasm mainly contains RanGDP (Figure 2C) [81]. This compartmentalization results from the asymmetric distribution of the regulators of the protein [81]. Namely, the cytoplasm contains Ran GTPase-activating protein (RanGAP) which will activate Ran to hydrolyse GTP into GDP, making RanGTP scares in the cytoplasm [81]. Conversely, the nucleus mainly contains guanine nucleotide exchange factors (RanGEFs) which will exchange GDP for GTP.

#### 4.3.2. Function

The RanGTPase system does not directly aid NTRs to cross the NPC, but regulates the loading and unloading of cargo form the NTRs on different sides of the nuclear membrane (Figure 2) [81]. This allows transport to occur against concentration gradients.

Upon association with RanGTP, conformational changes of exportin 1 will allow binding to the NES from its cargo (Figure 2A) [82]. Once arrived in the cytoplasm, binding by RanGAP induces GTP hydrolysis and causes the exportin-cargo complex to dissociate [82].

Nuclear import is essentially the opposite process, with release of cargo in the nucleus by association with RanGTP (Figure 2B) [65]. More specifically, importin-β1 has a low affinity for RanGDP, allowing it to form a heterodimer with importin-α in the cytoplasm [65]. Once bound, importin-α recognizes the cNLS of the cargo. This results in the formation of an importin β-importin α-cargo complex [65]. When this ternary complex enters the nucleus, RanGTP binds importin-β1, causing it to release importin-α, which subsequently releases its associated cargo [70]. Similarly, transportin 1 binds its cargo in the cytoplasm due to a low affinity for RanGDP (Figure 2B) [73]. Once in the nucleus, binding to RanGTP induces conformational changes and cargo release.

## 5. Evidence for Defective Nucleocytoplasmic Transport in ALS

### 5.1. Aberrant Subcellular Localization of Proteins

#### 5.1.1. Mislocalization of Nucleocytoplasmic Transport Proteins

A commonly used argument for the involvement of impaired nucleocytoplasmic transport in ALS is the mislocalization and aggregation of proteins of the nucleocytoplasmic transport machinery in postmortem brain and spinal cord tissues. Multiple studies observed aberrant nuclear morphologies, cellular mislocalization and/or aggregation of several nuclear transport receptors, nups or Ran-gradient proteins (Table 1). These include RanGap [83], importin-β1 [84,85], nup107 [83], nup62 [84,85] and gp210 [86].

Cellular mislocalization and/or aggregation of nucleocytoplasmic transport proteins has also been observed in multiple ALS models, both in vitro and in vivo (Table 2). For example, abnormal staining patterns of nup107 were observed in a (G_4_C_2_)_58_-*Drosophila* model [35] and in spinal cord motor neurons of a mutant SOD1 mouse model [86]. Interestingly, a reduction of eight nucleoporins was observed in mutant *C9orf72* iPSC-derived nuclei [89]. These include nup50, tpr, nup98, gp210, ndc1, nup107, nup133 and pom121. The authors proposed that a G_4_C_2_-repeat RNA induces a reduction in pom121, which initiates a decrease in the expression of the seven additional nucleoporins. This ultimately affects the localization of Ran and makes the cells sensitive for cellular toxicity [89]. Recently, the injury of the nuclear pore complexes in familial and sporadic ALS was linked to an increased nuclear accumulation of CHMP7, an important mediator of the quality control of the nuclear pore complex, using iPSC-derived motor neurons and postmortem tissue [91]. Inhibiting the nuclear export of CHMP7 triggered nup reduction as well as TDP-43 dysfunction, suggesting that CHMP7 could be a new therapeutic target [91]. The reason why CHMP7 starts to accumulate in the nucleus of motor neurons from ALS patients is still an open question.

Although these findings are all very suggestive for the involvement of nucleocytoplasmic-transport deficits, it is often unclear whether the observed abnormalities are sufficient to disturb nucleocytoplasmic transport. First, some nups are rather stationary, e.g., nup107 and nup205, and do not exchange from assembled nuclear pore complexes in non-dividing cells [96]. In addition, nups can have, besides their localization at nuclear pore complexes, a large cytoplasmic and nuclear pool. Therefore, it is possible that nups are sequestered outside the context of assembled nuclear pore complexes and it is unclear what the functional consequences are of their aggregate formation. To claim nucleocytoplasmic transport dysfunctions, it is also essential to show a depletion from the nuclear pore complex. Second, the findings are not always conclusive. For example, abnormal lamin-β staining was observed in fibroblasts derived from mutant *C9orf72* patients [90], but not in mutant *C9orf72* patient-derived induced neurons [92], spinal neurons derived from induced pluripotent stem cells (iPSCs) [97] or postmortem material from mutant C9orf72 patients [97]. Moreover, the aberrant localizations of nucleocytoplasmic transport proteins are not observed in all studies [87,88,90]. Third, the current knowledge is rather limited as no comprehensive analysis of all nuclear transport receptors and nups has been conducted so far. Fourth, postmortem analysis often gives us insights into end stage disease processes. As such, it is difficult to conclude whether the observed nucleocytoplasmic transport abnormalities are a driving force or a downstream by-product of disease progression. Lastly, two independent publications suggest that brightfield images of nuclear membrane proteins should be interpreted with caution [87,98]. They observed that the aberrant nuclear staining of RanGAP or importin-β1 could be misinterpreted and originate from the surface of shrunken nuclei. Altogether, this indicates that one should be cautious to link the observed postmortem data directly to nucleocytoplasmic-transport deficits as a causal factor of ALS pathology.

#### 5.1.2. Mislocalization of Cargoes

Despite the fact that genetic mutations in the gene encoding TDP-43 only account for approximately 4% of familial ALS, the vast majority of ALS cases exhibit cytoplasmic TDP-43 pathology. This seems to be due to a disturbance of its autoregulation, phase transition and nucleocytoplasmic transport, which are all part of an intrinsic control system regulating the physiological levels and localization of TDP-43 (for a review: [10]). Impaired nuclear import of TDP-43 seems to be a pathological feature common to many forms of ALS. In addition, many of the ALS-associated mutations in FUS are located in the NLS [12], indicating a prominent link between disease-associated protein mislocalization and nucleocytoplasmic transport.

TDP-43 aggregates have not only been suggested to be a consequence of failed nucleocytoplasmic transport, but also a cause [93]. Overexpressed TDP-43 C-terminal fragments (TDP-43 CTF) either co-aggregated with nups or induced their mislocalization [90]. This suggests that cytoplasmic TDP-43 inclusions have the capacity to disrupt the nuclear pore complex and its components, creating a self-feeding mechanism that exacerbates neurodegeneration. Indeed, cytoplasmic protein aggregation using artificial β-sheet proteins was shown to sequester and mislocalize proteins involved in nucleocytoplasmic transport [93].

Recently, mutant FUS was shown to be responsible for a disturbance of the Ran gradient as well as to affect the nuclear pore density in iPSC-derived spinal neurons containing FUS mutations [99]. Interestingly, an interaction of FUS with nup62 was observed and this protein also affected the liquid-liquid phase separation behavior of FUS in a cell-free system. Moreover, downregulation of nup62 had a positive effect on the phenotype in a mutant FUS *Drosophila* model [99].

### 5.2. Nucleocytoplasmic Transport Proteins as Modifiers of Disease

The observation that a loss of function (LOF, e.g., by transient knockdown) or a gain of function (GOF, e.g., by overexpression) of nucleocytoplasmic transport machinery proteins enhanced or reduced toxicity in ALS-models was an important factor in connecting nucleocytoplasmic transport to ALS pathology (Table 3). For example, a loss of function of importin-β1 enhanced toxicity in a poly-GR [41] and a poly-PR [37] *Drosophila* model. Similarly, loss of function of TPNO1 has been shown to enhance poly-GR [41] and poly-PR [35,37,92] induced toxicity. This suggests that impeded import plays a role in poly-PR and poly-GR induced pathology. 

However, these conclusions are not always convincingly supported by the experimental data. First, the complexity of the nucleocytoplasmic transport machinery makes it hard to decode underlying patterns found in the different screens. Knockdown of one protein can influence other transport proteins. For example, loss of nup98 has been shown to inhibit assembly of nup62 [103]. Loss of nup62 enhanced toxicity [37], but loss of nup98 suppressed toxicity [35,90]. As a second example, one study concluded that loss of function of importin-β1 enhanced poly-GR induced toxicity, but importin-α1 suppressed this toxicity [41]. This might be surprising as importin-β1 mediates classical import via importin-α. In addition, other importins showed the opposite phenotype including importin 4, importin 5 and importin 7 (Table 3). The reasons for these discrepancies are currently unknown, but a possible explanation might be that the cargoes are imported via importin-β1, independently of importin-α. Another possibility is that other import pathways are favored upon loss of importin-α1. Moreover, nuclear transport receptors and nups fulfill additional functions in addition to nucleocytoplasmic transport. These functions could offer a so far underestimated and potentially important explanation of the observed modifier effects.

Second, the vast amount of current knowledge about nuclear pore complex assembly is based on data gathered in dividing cells. This could be misleading. For example, knock down of nup107—which has repeatedly been observed as a suppressor [35,37,90]—reduced the formation of nuclear pore complexes [104]. This could suggest that reduced nucleocytoplasmic transport is beneficial. However, the assembly of nup107 in post-mitotic nuclear pore complexes has been shown to be extremely stable [105]. Hence, until more data are obtained in post-mitotic cells, it is difficult to conclude that nup107 knockdown indeed influences assembled nuclear pore complexes in these models.

### 5.3. Functional Assays

Additional arguments to suggest nucleocytoplasmic-transport deficits in ALS were provided by studies with functional assays that measure nucleocytoplasmic transport in ALS models (Table 4). These assays are mainly based on analyzing the nuclear and cytoplasmic localization of a shuttling fluorophore that contains an NLS and NES (for a review: [106]). For instance, Zhang et al. concluded that there was significantly reduced transport of an NLS-NES-GFP reporter in mutant *C9orf72* iPSC-derived motor neurons based on FRAP experiments [83]. In addition, Chou et al. observed that the nuclear/cytoplasmic ratio of the reporter NES-tdTomato-NLS was significantly reduced in fibroblasts of mutant *C9orf72*, mutant *TARDBP* and sporadic ALS patients [90]. Recently, a similar observation was made in iPSC-derived spinal neurons containing a FUS mutation (M511Nfs) [99]. Fluorescence recovery after photobleaching (FRAP) of the nucleus was lower in the mutant FUS neurons. Altogether, these data suggest that nucleocytoplasmic-transport deficits could play an important role in ALS pathology, although one has to be careful with the interpretation of results obtained from FRAP experiments [106].

Moreover, there are some inconsistencies especially in relation to the effect of the DPRs. For instance, one study suggested that PR_20_ peptides could block the nuclear pore complex [45], while another study found that adding PR_20_ peptides accelerated passive nuclear efflux [47]. Hayes et al. also observed a negative impact of poly-PR and poly-GR peptides on TPNO1-related import [47]. Conversely, no TPNO1-import deficits were observed by Khosravi et al. who made use of poly-PR/GR expressing cells [100].

Recently, we confirmed the absence of a direct effect of poly-PR and poly-GR in different cell systems and using different reporter constructs imported by different nuclear transport receptors [108,109]. We did not observe impeded XPO1-mediated export, importin-mediated import or TPNO1-mediated import in the presence of poly-PR and poly-GR in HeLa cells. Moreover, no decreased transport was induced by poly-PR and poly-GR in neuronal-like SH-SY5Y cells and in iPSC-derived motor neurons, excluding the involvement of cell-type specific effects and again strongly indicating that poly-GR or poly-PR do not directly induce an impediment of nucleocytoplasmic transport by physically blocking the nuclear pore [108].

As many transport-related proteins have also been shown to localize at stress granules, including nuclear transport factors and nucleoporins [77,107,110,111,112,113], recruitment of transport proteins into stress granules could lead to a general decrease in nucleocytoplasmic transport [107]. While our initial data were in line with this hypothesis [108], overall stress seems to be the cause of the decrease in nucleocytoplasmic transport rather than the formation of stress granules [114].

## 6. Therapeutic Potential

The observation that altering nucleocytoplasmic transport proteins can influence ALS-related toxicity is of great interest, in particular for future therapeutic opportunities. For example, beneficial effects were observed upon exportin 1 downregulation (Table 3) [83,90,102]. One of these studies made use of a *Drosophila* model, in which knock down of exportin 1 decreased (G_4_C_2_)_30_-repeat-induced toxicity [83]. It was hypothesized that reducing nuclear export slowed down neurodegeneration by compensating for impaired nuclear import. This was confirmed via the use of a selective inhibitor of nuclear export (SINE) compound, namely KPT-276 [83]. Biogen is currently testing a similar SINE compound in a phase 1 safety trial [115]. However, not all studies agree on the beneficial effects of exportin 1 LOF (Table 3) [35,37,41,90]. For instance, exportin 1 downregulation enhanced toxicity in another *Drosophila* model expressing 58 repeats of G_4_C_2_ [35]. This was also confirmed by an exportin 1 inhibitor, namely leptomycin B (LMB). One possible explanation is the detectable presence of DPRs in the latter model, which is thought to be absent in the first model. Indeed, downregulation of exportin 1 also enhanced toxicity in a pure poly-PR *Drosophila* model [37].

A second therapeutic strategy is focused on enhancing nuclear import. Indeed, downregulation of importin-β1 [37,41] or TPNO1 [35,37,41] enhanced, while overexpression of TPNO1 suppressed [92] mutant *C9orf72* associated toxicity (Table 3).

Another exciting opportunity is related to the chaperoning function of importins. Remarkably, multiple studies observed the ability of TPNO1 and importin-β1 to prevent or even revert aberrant phase transitions of their cargoes, including TDP-43 and FUS [77,78,79,80]. It is thought that elevating their expression could have therapeutic potential to restore homeostasis of these RNA binding proteins and to diminish neurodegeneration [116].

## 7. Nucleocytoplasmic Transport and the Aging Brain

While it is generally accepted that a complex interplay between genes, environment and aging is the cause of ALS, there is currently no clear evidence about which environmental factors are potentially involved. However, aging indeed seems to increase the risk to develop ALS, at least to a certain extent. Prominent among the many cellular processes that decline during aging is nucleocytoplasmic transport [63], with the strongest evidence indicating alterations in the integrity of the nuclear pore complex. In dividing cells, nuclear pore complexes disassemble during nuclear membrane breakdown in mitosis and subsequently reassemble with newly synthesized proteins [117]. However, the disassembly of entire nuclear pore complexes might not be possible in post-mitotic cells, as this could lead to partial nuclear membrane breakdown. Instead, individual subcomplexes have to be exchanged over time, with some experiencing a fast turnover while others are extremely stable. Dynamic peripheral nuclear pore components (e.g., nup62, nup153 and nup50) are continuously exchanged, whereas scaffold nups (e.g., nup107, nup93) are suggested to be extremely long-lived [59,118]. Remarkably, approximately 25% of the core scaffold component nup205, was not replaced in rat brains after one year [118]. These long-lived proteins might be particularly vulnerable to protein damage over time. Consistent with this idea, it was found that nup93, a linker nup, suffers from oxidative damage in old cells [59]. Subsequently, loss of nup93 could result in a loss of FG-nups and a deterioration of the permeability barrier. Indeed, it was found using dextran influx assays that the nuclear pore complex became leaky with age [105]. In conclusion, a subset of scaffolding nups does not turn over in post-mitotic cells, making them particularly sensitive to oxidative damage over time. This could lead to an age-related deterioration of nuclear pore complexes with leaky nuclei as a result.

Two additional studies demonstrated the failure of nucleocytoplasmic transport during physiological aging. Mertens et al. made use of induced neurons (iNs), which are directly converted from human fibroblasts and, as such, retained an aging-associated gene expression signature [119]. The authors observed that these iNs displayed an age-related decline in the expression of the importin RanBP17 and a concomitant impaired nucleocytoplasmic compartmentalization [119]. Similarly, Pujol et al. observed an age-related decrease in the protein concentration of transport factors in human fibroblast (CAS, importin-α1 and RanBP1, but not importin-β1) and a decrease in protein import in fibroblasts from old donors [120].

Taken together, these studies suggest that the nucleocytoplasmic transport machinery is a target of aging, as both an age-related deterioration of nuclear pore complexes and a change in expression of transport proteins have been observed. This could provide an important link between aging and neurodegenerative diseases, such as ALS.

## 8. Conclusions

In recent years, several lines of evidence indicated the involvement of nucleocytoplasmic-transport deficits in ALS pathology, particularly in mutant *C9orf72*-ALS. However, the current evidence is often circumstantial and does not clarify the exact causal and temporal role of nucleocytoplasmic transport deficits in mutant *C9orf72* patients. Gaining this knowledge will be of great importance if we want to target therapeutically nucleocytoplasmic transport in general and/or specific proteins involved in this process to slow down or stop the neurodegeneration leading to ALS.

## Figures and Tables

**Figure 1 ijms-22-12175-f001:**
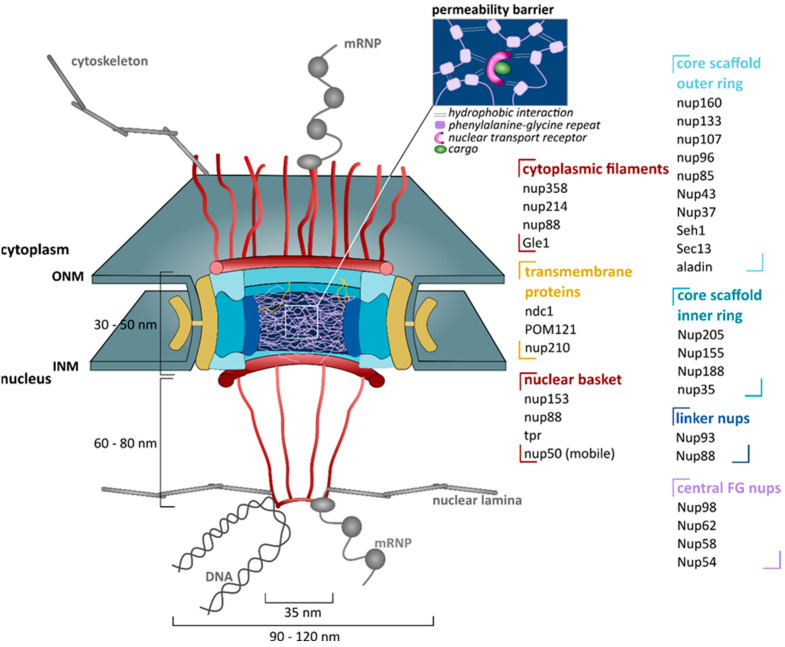
Structural organization of the nuclear pore complex. Nuclear pore complexes (NPCs) reside in circular openings in the nuclear envelope where the outer and inner nuclear membrane (ONM and INM, respectively) fuse. An NPC consists of different proteins called nucleoporins (nups), which are grouped in subcomplexes. Transmembrane nucleoporins anchor the NPC in the nuclear membrane. They connect to the core scaffold that exists of an outer ring and an inner ring. Linker nucleoporins anchor phenylalanine-glycine nucleoporins (FG nups) to the core scaffold, such that they line and fill the central tube. There, they phase separate into a dense polymer meshwork, which forms a highly selective permeability barrier. Peripheral structures consist of cytoplasmic filaments and a nuclear basket. Cytoplasmic filaments connect to the cytoskeleton and to the protein synthesis machinery. The nuclear basket connects the NPC to aspects of nuclear metabolism, such as mRNA biogenesis and genome organization. ndc1: nuclear division cycle protein 1, pom121: pore membrane protein 121, tpr: translocated promoter region, seh1: Sec13 homologue 1. NPC structure is based on the reviews of Strambio-De-Castillia et al. [61] and Juhlen and Fahrenkrog [62]. Image of the permeability barrier is based on the review of Kim and Taylor [63].

**Figure 2 ijms-22-12175-f002:**
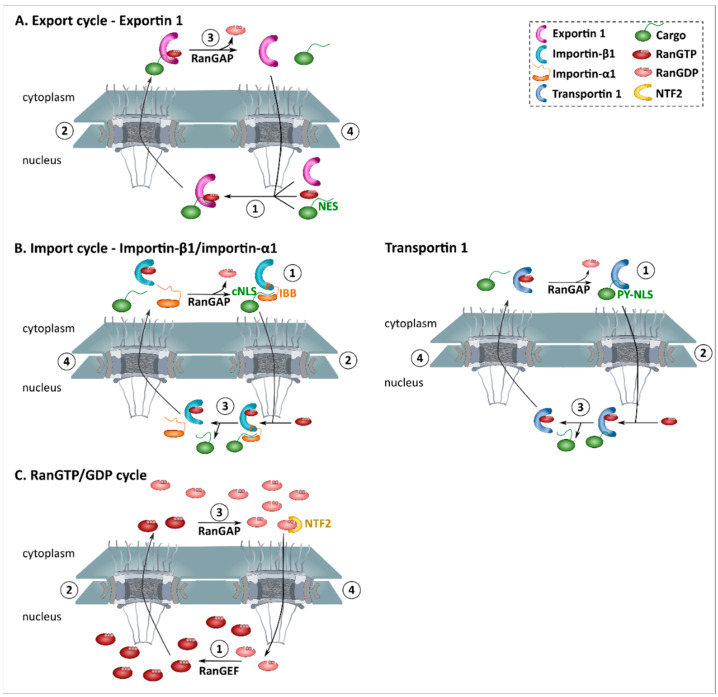
Schematic representation of Ran-dependent nucleocytoplasmic transport. Transport across the NPC is achieved by nuclear transport receptors, importins and exportins, and powered by a RanGTP gradient. (**A**) Nuclear export. (**1**) In the presence of high RanGTP concentrations, exportin 1 is able to bind cargoes containing a nuclear export signal (NES). (**2**). As exportin 1 has the unique ability to interact with the FG-repeats of the central channel, the exportin-cargo complex undergoes rapid passage through the pore. (**3**) In the cytoplasm, hydrolysis of RanGTP to RanGDP will cause the release of the cargo. (**4**) Exportin 1 is recycled back to the nucleus. (**B**) Nuclear import. (**1**) Due to the low affinity for RanGDP, importins bind their cargo in the cytoplasm. Left panel: a protein containing a classical nuclear localization signal (cNLS) is bound by the adaptor importin-α, which subsequently binds importin-β1 via its importin-β-binding domain (IBB). Right panel: proteins containing proline-tyrosine-nuclear localization signal (PY-NLS) are recognized by transportin 1. (**2**) Importins can transiently open the FG-meshwork of the central channel, allowing the passage of bound cargo. (**3**) In the nucleus, binding of RanGTP induces conformational changes, which induces cargo release. (**4**) Importins are exported back to the cytoplasm. (**C**) RanGTP/GDP cycle. (**1**) The high concentrations of RanGTP (Ran guanosine triphosphate) in the nucleus is maintained by RanGEF (Ran—guanine exchange factor). (**2**) RanGTP undergoes a continuous efflux out the nucleus via its binding to NTRs (see A and B). (**3**) RanGAP (Ran—GTPase activating protein) mediated hydrolysis of RanGTP into RanGDP, resulting in high concentrations of RanGDP (Ran guanosine diphosphate) in the cytoplasm. (**4**) RanGDP is imported into to the nucleus by NTF2 (nuclear transport factor 2) to restore depleted nuclear levels of RanGTP. Image is based on the review of Cook et al. [65].

**Table 1 ijms-22-12175-t001:** Aberrant localization of proteins related to nucleocytoplasmic transport in postmortem tissue. Overview of the available evidence indicating whether or not proteins involved in nucleocytoplasmic transport were abnormally distributed in ALS and/or FTD patients. sALS: sporadic ALS; sFTD: sporadic frontotemporal dementia.

Proteins	Observations	Models	Refs
**Importins**			
**Importin-** **β1**	Irregular, disrupted at nuclear membrane, increased cytoplasmic localization	sALS, spinal cord	[84]
	Nuclear depletion	sALS, spinal cord	[85]
	No abnormalities	mutant *C9orf72* ALS	[87]
**Importin-** **α1**	Nuclear depletion and cytoplasmic accumulation	mutant *C9orf72* and sFTD	[88]
**Importin-** **α4**	Nuclear depletion and cytoplasmic accumulation	mutant *C9orf72* and sFTD	[88]
**Nups—TM**			
**Gp210**	Abnormal nuclear precipitations and cytoplasmic upregulation	sALS, spinal cord	[86]
**Pom121**	Reduced expression (together with nup50, tpr, nup98, ndr1, nup107, nup133)	mutant *C9orf72*, motor cortex and spinal cord	[89]
**Nups—scaffolds**			
**Nup107**	Aggregates at nuclear membrane	mutant *C9orf72* ALS, motor cortex	[83]
**Nup205**	Abnormal nuclear localization	mutant *C9orf72* ALS, motor cortex	[83]
	Loss of immunoreactivity and large cytoplasmic inclusions	mutant TDP-43 + sALS, motor cortex	[90]
	Abnormal perinuclear punctate staining	mutant *C9orf72*, motor cortex	[90]
	No abnormalities	mutant *SOD1*, motor cortex	[90]
**Nups—linkers**			
**Nup88**	Tortuous and redundant nuclear contours	mutant *SOD1* ALS + sALS, spinal cord	[85]
**Nups—central channel**			
**Nup62**	Irregular disrupted nuclear membrane, aggregates, nuclear localization	sALS, spinal cord	[84]
	Tortuous and redundant nuclear contours	mutant *SOD1* + sALS, spinal cord	[85]
**Nup—outer structures**			
**Nup50**	Abnormal nuclear precipitations and cytoplasmic upregulation	sALS, spinal cord	[86]
**Ran gradient**			
**RanGap**	Abnormal nuclear localization	mutant *C9orf72* ALS, motor cortex	[83]
	Abnormal nuclear precipitations and cytoplasmic upregulation	SALS, spinal cord	[86]
	No abnormalities	mutant TDP-43 + mutant *C9orf72* + SALS motor cortex	[90]
	No abnormalities	mutant *C9orf72* ALS	[87]

**Table 2 ijms-22-12175-t002:** Aberrant localization of proteins related to nucleocytoplasmic transport in ALS models. Overview of the data indicating whether or not proteins involved in nucleocytoplasmic transport were affected in in vitro and in vivo models for different forms of ALS. iPSC: induced pluripotent stem cell; CTF: C-terminal fragment.

**In Vitro**			
**Proteins**	**Observations**	**Models**	**Refs**
**Importins**			
**TPNO3**	No abnormal staining	Patient-derived mutant *C9orf72* induced neurons	[92]
**Importin-α3**	No abnormal staining	Patient-derived mutant *C9orf72* induced neurons	[92]
**Exportins**			
**Exportin 5**	No abnormal staining	Patient-derived mutant *C9orf72* induced neurons	[92]
**Nups**			
**Mab414 (nup62, nup153›, nup214, nup358)**	Abnormal staining	Mutant TDP-43 iPSC motor neurons	[90]
**NPC proteins**	Interaction with PR_50_ and GR_50_	HEK293T cells expressing PR_50_ or GR_50_	[41]
**Gle1, nup88, nup214, nup358, nup35, nup93, nup58, nup62, nup98, nup107, nup155, nup160 nup205, nup153, aladin, nxf1**	Aggregation together with TDP-43 CTF	BioID approach in Neuro-2a cells expressing TDP-43 CTF	[90]
**Nups—TM**			
**Pom121**	Mislocalization upon TDP-CTF OE	Neuro-2a cells cotransfected with plasmid expressing pom121 and TDP-43 CTF	[90]
	Reduced expression	Patient-derived mutant *C9orf72* iPSC induced neurons	[89]
**Gp210**	Mislocalization upon TDP-CTF OE	Neuro-2a cells cotransfected with plasmid expressing gp210 and TDP-43 CTF	[90]
**Nups—scaffolds**			
**Nup205**	Abnormal staining	Mutant TDP-43 and mutant *C9orf72* patient fibroblasts	[90]
	No abnormal staining	Sporadic ALS patient fibroblasts	[90]
**Nups—central channel**			
**Nup62**	Irregular nuclear contour	Spinal cord of SOD1^G93A^ mouse model	[85]
**Nup98**	Disturbed distribution	Neuro-2a cells transfected with TDP-43 CTF	[90]
**Lamina**			
**Lamina B1**	Abnormal staining	Mutant TDP-43 and mutant *C9orf72* patient fibroblasts	[90]
	No abnormal staining	Sporadic patient fibroblasts	[90]
	Abnormal staining	Mutant TDP-43 iPSC motor neurons	[90]
	Abnormal staining	Mouse primary cortical neurons expressing TDP-43 CTF/TDP-43^Q331K^/TDP-43^M337V^	[90]
	No abnormal staining	Patient-derived mutant *C9orf72* induced neurons	[92]
**Ran gradient**			
**RanGAP**	Nuclear puncta	S2 *Drosophila* cells transfected with (G_4_C_2_)_30_	[83]
	Nuclear puncta	Mutant *C9orf72* iPSC-derived neurons	[83]
	Abnormal staining	Mouse primary cortical neurons expressing TDP 43 CTF or TDP-43-mtNLS	[90]
	No abnormal staining	Mouse primary cortical neurons expressing TDP 43^Q331K^ or TDP 43^M337V^	[90]
	No abnormal staining	Patient-derived mutant *C9orf72* induced neurons	[92]
**RanGEF**	Nuclear depletion	Patient derived mutant *C9orf72* induced neurons	[92]
**Ran**	Reduced nuclear/cytoplasmic ratio	S2 *Drosophila* cells transfected with (G_4_C_2_)_30_	[83]
	Reduced nuclear/cytoplasmic ratio	Patient-derived mutant *C9orf72* induced neurons	[83]
	No change in nuclear/cytopIasmic ratio	Mouse primary cortical neurons expressing TDP-43 CTF or TDP-43-mtNLS	[90]
**RAN export**			
**Thoc2**	Cytoplasmic mislocalization	HEK293T cells expressing TDP-43 fragments	[93]
**In vivo**			
**Proteins**	**Observations**	**Models**	**Refs**
**Importins**			
**TPNO1**	Cytoplasmic aggregates	Prp-TDP-43^A315T^-GFP mouse model	[94]
**Importin-α1**	Cytoplasmic accumulation and nuclear depletion	Salivary gland cells of *Drosophila* expressing poly-GA_64_ or poly-GR_64_	[88]
	Cytoplasmic accumulation and nuclear depletion	Salivary gland cells of *Drosophila* expressing ΔNLS-TDP-43, human TDP-43 or TDP-43^Q331K^	[88]
	Cytoplasmic accumulation and nuclear depletion	Salivary gland cells of *Drosophila* expressing poly-GA_64_ or poly-GR_64_	[88]
**Importin-α4**	Cytoplasmic accumulation and nuclear depletion	Salivary gland cells of *Drosophila* expressing ΔNLS-TDP-43, human TDP-43 or TDP-43^Q331K^	[88]
**Nups**			
**Mab414**	No abnormal staining	Salivary gland cells of *Drosophila* expressing poly-GA_64_ or poly-GR_64_	[88]
**(nup62, nup153, nup214, nup358)**	No abnormal staining	Salivary gland cells of *Drosophila* expressing ΔNLS-TDP-43, human TDP-43 or TDP-43^Q331K^	[88]
**Nups—TM**			
**Gp210**	Increased nuclear localization	Spinal cord motor neurons of SOD1^G93A^ mouse model	[86]
**Nups—Scaffolds**			
**Nup107**	Inclusions near nuclear envelope	Salivary gland cells of *Drosophil**a* expressing (G_4_C)_58_	[35]
	Increased nuclear localization	Spinal cord motor neurons of SOD1^G93A^ mouse model	[86]
**Nup205**	Increased nuclear localization	Spinal cord motor neurons of SOD1^G93A^ mouse model	[86]
**Nups—outer structures**			
**Nup50**	No abnormal staining	Salivary gland cells of *Drosophila* expressing poly-GA _64_ or poly-GR_64_	[88]
**Lamina**			
**Lamina C**	Abnormal nuclear membrane	Salivary gland cells of *Drosophila* expressing (G_4_C_2_)_58_	[35]
**Ran gradient**			
**RanGap**	Increased nuclear localization	Spinal cord motor neurons of SOD1^G93A^ mouse model	[86]
	No abnormal staining	Salivary gland cells of *Drosophila* expressing poly-GA_64_ or poly-GR_64_	[88]
	Increased nuclear localization	PrP-TDP43^A315T^ GFP mouse model	[94]
	Increased localization to nuclear invaginations	Mice expressing (G_4_C_2_)_149_ via intracerebroventricular injections with AAV	[95]

**Table 3 ijms-22-12175-t003:** Modifiers of toxicity. Summary of all the ALS-related models in which the effect of changing nucleocytoplasmic transport proteins on the different phenotypes was determined. S indicates suppressors. A protein is defined as a suppressor when a loss of function (LOF) suppresses the phenotype, a gain of function (GOF) enhances the phenotype or overexpression (OE) enhances the phenotype. E indicates an enhancer. A protein is defined as an enhancer when a LOF enhances the phenotype, a GOF suppresses the phenotype or OE suppresses the phenotype.

Proteins	E/S	Observations	Models	Refs
**Importins**				
**Importin-** **β** **1**	E	LOF enhances phenotype	*Drosophila* expressing poly-GR_50_	[41]
	E	LOF enhances phenotype	*Drosophila* expressing poly-PR_25_	[37]
**Importin-α1**	S	OE enhances phenotype	*Drosophila* (G_4_C_2_)_30_—no DPRs detected	[83]
	S	LOF suppresses phenotype	*Drosophila* expressing poly-GR_50_	[41]
**Importin-α3**	E	OE suppresses phenotype	Yeast expressing poly-PR_50_	[92]
	E	LOF enhances phenotype	*Drosophila* expressing poly-PR_25_	[37]
	E	OE suppresses NC-transport phenotype	HeLa cells expressing GA_149_	[100]
**Importin-α4**	E	LOF enhances phenotype	*Drosophila* expressing poly-GR_50_	[41]
	E	LOF enhances phenotype	*Drosophila* expressing poly-PR_25_	[37]
	E	OE suppresses NC-transport phenotype	HeLa cells expressing GA_149_	[100]
**TPNO1**	E	LOF enhances phenotype	*Drosophila* (G_4_C_2_)_58_—DPRs detected	[35]
	E	OE suppresses phenotype	Yeast expressing poly-PR_50_	[92]
	E	LOF enhances phenotype	*Drosophila* expressing poly-GR_50_	[41]
	E	LOF enhances phenotype	*Drosophila* expressing poly-PR_25_	[37]
**TPNO3**	E	OE suppresses phenotype	Yeast expressing poly-PR_50_	[92]
**Importin 4**	S	LOF suppresses phenotype	*Drosophila* expressing poly-PR_25_	[37]
**Importin 5**	S	LOF suppresses phenotype	*Drosophila* expressing poly-GR_50_	[41]
**Importin 7**	S	LOF suppresses phenotype	*Drosophila* expressing poly-GR_50_	[41]
**Importin 9**	E	OE suppresses phenotype	Yeast expressing poly-PR_50_	[92]
**Importin 11**	E	OE suppresses phenotype	Yeast expressing poly-PR_50_	[92]
	E	LOF enhances phenotype	*Drosophila* expressing poly-PR_25_	[37]
**Exportins**				
**Exportin 1**	S	LOF suppresses phenotype–confirmed by KPT-276	*Drosophila* (G_4_C_2_)_30_—no DPRs detected	[83]
	S	LOF suppresses phenotype	*Drosophila* expressing human FUS^R521G^	[101]
	E	LOF enhances phenotype—confirmed by LMB	*Drosophila* (G_4_C_2_)_58_ - DPRs detected	[35]
	E	LOF enhances phenotype	*Drosophila* expressing poly-GR_50_	[41]
	E	LOF enhances phenotype	*Drosophila* expressing poly-PR_25_	[37]
	S	2.5 nM KPT-350/KPT-335 is mildly protective (no impaired transport)	Rat cortical neurons OE human TDP-43^WT^	[102]
	E	10 nM KPT-350/KPT-335 is toxic	Rat cortical neurons OE human TDP-43^WT^	[102]
		KPT-350 induces a partial rescue limited by weight loss	Rat model AAV9 brain injected TDP-43 mRNA	[102]
	S	KPT-335 treatment suppresses phenotype	Mouse cortical neurons expressing TDP-43 CTF or TDP-43^Q331K^	[90]
	S	1 µM KPT-335/KPT-276 treatment suppresses phenotype	*Drosophila* TDP-43^WT^ or TDP-43^Q331K^ OE	[90]
	E	150 nM KPT335/KPT-276 treatment enhances phenotype	Mouse cortical neurons expressing TDP-43 CTF or TDP-43^Q331K^	[90]
	E	5 µM KPT-335/KPT-276	*Drosophila* TDP-43^WT^ or TDP-43^Q331K^ OE	[90]
**Exportin 5**	E	OE suppresses phenotype	Yeast expressing poly-PR_50_	[92]
**Nups—TM**				
**Ndc1**	S	OE enhances phenotype	Yeast expressing poly-PR_50_	[92]
**Nups—scaffolds**				
**Nup107**	S	LOF suppresses phenotype	*Drosophila* (G_4_C_2_)_58_ - DPRs detected	[35]
	S	LOF suppresses phenotype	*Drosophila* TDP-43^WT^ or TDP-43^Q331K^ OE	[90]
	S	LOF suppresses phenotype	*Drosophila* expressing poly-PR_25_	[37]
**Nup160**	S	LOF suppresses phenotype	*Drosophila* (G_4_C_2_)_58_ - DPRs detected	[35]
**Nup205**	S	LOF suppresses phenotype	*Drosophila* expressing poly-GR_50_	[41]
**Seh1**	E	LOF enhances phenotype	*Drosophila* expressing poly-PR_25_	[37]
**Nup155**	S	LOF suppresses phenotype	*Drosophila* expressing poly-PR_25_	[37]
	S	LOF suppresses phenotype	*Drosophila* expressing human FUS^R521G^	[101]
**Nups—linkers**				
**Nup93**	S	LOF suppresses phenotype	*Drosophila* TDP-43^WT^ or TDP-43^Q331K^ OE	[90]
	E	LOF enhances phenotype	*Drosophila* expressing poly-PR_25_	[37]
**Nups—central channel**				
**Nup96—Nup98**	S	LOF suppresses phenotype	*Drosophila* (G_4_C_2_)_58_ - DPRs detected	[35]
	S	LOF suppresses phenotype	*Drosophila* TDP-43^WT^ or TDP-43^Q331K^ OE	[90]
	S	LOF suppresses phenotype	*Drosophila* FUS^WT^, FUS^R518K^, FUS^R521C^ OE	[99]
**Nup62**	E	LOF enhances phenotype	*Drosophila* expressing poly-PR_25_	[37]
	S	LOF suppresses phenotype	*Drosophila* FUS^WT^, FUS^R518K^, FUS^R521C^ OE	[99]
	S	OE enhances phenotype	*Drosophila* FUS^WT^, FUS^R518K^, FUS^R521C^ OE	[99]
	E	OE suppresses NC-transport phenotype	HeLa cells expressing GA_149_	[100]
**Nup54**	E	OE suppresses NC-transport phenotype	HeLa cells expressing GA_149_	[100]
	S	LOF suppresses phenotype	*Drosophila* FUS^WT^, FUS^R518K^, FUS^R521C^ OE	[99]
**Nups—outer structures**				
**Nup50**	E	LOF enhances phenotype	*Drosophila* (G_4_C_2_)_58_ - DPRs detected	[35]
	S	LOF suppresses phenotype	*Drosophila* expressing poly-PR_25_	[37]
**Nup153**	E	LOF enhances phenotype	*Drosophila* (G_4_C_2_)_58_—DPRs detected	[35]
**Nup214**	S	LOF suppresses phenotype	*Drosophila* TDP-43^WT^ or TDP-43^Q331K^ OE	[90]
	S	LOF suppresses phenotype	*Drosophila* FUS^WT^, FUS^R518K^, FUS^R521C^ OE	[99]
**Tpr**	E	LOF enhances phenotype	*Drosophila* expressing poly-PR_25_	[37]
**RanGradient**				
**RanGap**	E	GOF mutation/OE suppresses phenotype	*Drosophila* (G_4_C_2_)_30_—no DPRs detected	[83]
	E	LOF enhances phenotype	*Drosophila* expressing poly-PR_25_	[37]
**RanGEF**	S	OE enhances phenotype	*Drosophila* (G_4_C_2_)_30_—no DPRs detected	[83]
	S	OE enhances phenotype	Yeast expressing poly-PR_50_	[92]
	E	LOF enhances phenotype	*Drosophila* expressing poly-PR_25_	[37]
**Ran**	E	Dominant negative mutation enhances phenotype	*Drosophila* (G_4_C_2_)_58_—DPRs detected	[35]
**RAN export**				
**Alyref**	S	LOF suppresses phenotype	*Drosophila* (G_4_C_2_)_58_—DPRs detected	[35]
**Gle1**	E	LOF enhances phenotype	*Drosophila* (G_4_C_2_)_58_—DPRs detected	[35]

**Table 4 ijms-22-12175-t004:** Functional assays measuring different import pathways. Overview of the different assays used to measure the effects on nucleocytoplasmic transport. n.a.: not applicable.

Assay	Models	Observations	Suggested Mechanism	Refs
**Imp-β1/α1**				
**NLS-NES-GFP**	*Drosophila* salivary glands expressing (G_4_C_2_)_30_ repeats	Reduced N/C ratio	RanGap sequestration in nuclear RNA foci	[107]
	Mutant *C9orf72* iPSC-derived neurons	Reduced import based on FRAP		
**GFP-NLS-NES**	U2OS treated with PR_20_ peptides	Reduced import over time	Pores are blocked by poly-PR peptides	[45]
**BSA-NLS**	Digitonin-treated HeLa cells exposed to PR_20_ peptides	Reduced import		
**RFP-NLS_TDP43_**	HeLa cells transfected with GA_149_ expressing plasmid	Increased cytoplasmic levels	Cytoplasmic poly-GA aggregates	[100]
	HeLa cells transfected with GR_149_ expressing plasmid	No consistent differences observed	n.a.	
	HeLa cells transfected with PR_175_ expressing plasmid	No differences observed	n.a.	
**NES-tdTomato-NLS**	Primary mouse cortical neurons expressing TDP-43 CTF/mTDP-43	Reduced N/C ratio	TDP-43 aggregates sequester NC-transport proteins	[90]
	Fibroblasts of mutant *C9orf72*, TDP-43 and sALS patients			
	HEK293T cells transfected with PR_50_/GR_50_ expressing plasmids	Mislocalization of reporter	Stress granules sequester NC-transport proteins	[107]
	ALS-FUS human spinal neurons and isogenic controls	Decreased nuclear import	Increased interaction of mutant FUS with Nup62	[99]
**Artificial importin-** **β** **cargo based on FRET**	Permeabilized HeLa cells incubated with PR_20_ and GR_20_ peptides	Decreased nuclear import	Poly-PR and poly-GR bind and disrupt cargo loading of importin-β	[47]
**Fluorescent dextrans**	Permeabilized HeLa cells incubated with PR_20_ and GR_20_ peptides	Increased passive transport		
**Hormone-induced import assay:** **GCR_2_-GFP_2_-TDP43 or GCR_2_-GFP_2_-(MBP)-cNLS**	HeLa cells incubated with TMR-GR_25_ peptides	Reduced import	Reduced solubility of importin-α/β via poly-GR binding	[46]
	HeLa cells incubated with TMR-PR_25_ peptides	No difference observed	n.a.	
**NLSSV40-mNeonGreen2x-NES_pki_**	HeLa cells incubated with GR_20_ or PR_20_ +/− leptomycin B (LMB)	No differences observed	n.a.	[108]
	HeLa cells transduced with GR_100_ or PR_100_ +/− LMB	No differences observed	n.a.	
	SH-SY5Y cells transduced with GR_100_ or PR_100_ +/− LMB	No differences observed	n.a.	
	iPSC-derived MNs transduced with GR_100_ or PR_100_ +/− LMB	No differences observed	n.a.	
**NLSc-myc-GFP2x-NES_ikb2_**	HeLa cells transduced with GR_100_ or PR_100_	No differences observed	n.a.	
**TPNO**				
**RFP-NLS_pY(hnRNPA1)_**	HeLa cells transfected with GA_149_, GR_149_ or PR_175_ expressing plasmids	No differences observed	n.a.	[100]
**YFP-M9-CFP**	Digitonin treated HeLa cells incubated with PR20 and GR20 peptides	Decreased nuclear import	Poly-PR and poly-GR bind and disrupt cargo loading of TPNO1	[47]
**NES_pki_-mNeonGreen2x-NLS_FUS_**	HeLa cells incubated with GR_20_ or PR_20_	No differences observed	n.a.	[108]

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
