# Peer review of "The Role of Nucleocytoplasmic Transport Defects in Amyotrophic Lateral Sclerosis"

_ijms, 2021, doi:10.3390/ijms222212175_

Round 1
Reviewer 1 Report
This review by Vanneste and Van den Bosch explores the involvement of nucleocytoplasmic-transport defects in ALS pathogenesis. The topic is timely, and the authors have the expertise to write such a review. The reviewer really appreciates the work put into this review and provided comments highlighting minor concerns.
The authors should include a paragraph about rarer genetic causes of ALS as VAPB, UBQLN2 and OPTN among others.
A figure that shows the nucleocytoplasmic transport proteins and a short explanation of their role would be appreciated.
Table 2 should be properly named in the table legend.
The reference Xiao et al., 2015 should be included and discussed
Author Response
The authors should include a paragraph about rarer genetic causes of ALS as VAPB, UBQLN2 and OPTN among others.
Answer: We have mentioned in our revised version that there are indeed rare genetic forms of ALS (in the introduction of the genetic part, p. 4). However, as almost all the information documenting a potential role of nucleocytoplasmic transport defects in ALS is from models related to the four major genetic cases of fALS, we have focused on introducing these more general genetic causes.
A figure that shows the nucleocytoplasmic transport proteins and a short explanation of their role would be appreciated.
Answer: We have included an additional part in our review illustrating (using two new figures) and explaining as short as possible the (many) different aspects of nucleocytoplasmic transport. We agree with the reviewer that this will help to understand the impact of the (potential) nucleocytoplasmic defects described in our review.
Table 2 should be properly named in the table legend.
Answer: We thank the referee for this suggestion. We have included a short description of the content of each Table in the table legends.
The reference Xiao et al., 2015 should be included and discussed
Answer: We omitted this reference in our original version of the review as the observation that the short isoform of C9orf72 is localized at the nuclear membrane (which could suggest its involvement in nucleocytoplasmic transport) is debated as another study (Frick et al, 2018) which made use of knock-out validated antibodies, did not detect this isoform in the human CNS. We have included and discussed this information in our revised version. One explanation for this discrepancy could have to do with the problems which were observed with the quality of the commercially available antibodies directed against C9orf72.
- Frick, P et al. Novel antibodies reveal presynaptic localization of C9orf72 protein and reduced protein levels in C9orf72 mutation carriers. Acta Neuropathologica Communications 6, 72-72 (2018)..
Reviewer 2 Report
The work entitled "The Role of Nucleocytoplasmic Transport Defects in Amyotrophic Lateral Sclerosis" by Joni Vanneste and Ludo Van Den Bosch is a very good review article which covers the actual knowledge on defects related to nucleocytoplasmic transport in ALS. After a general introduction, the description of the main causes of ALS and its histopathology, authors collect evidence of the known defects related to nucleocytoplasmic transport in ALS.
The manuscript is well-organized and well-written. English language is correct and only fine/minor spell checks are required. Here few comments/required revisions:
- Tables help to sum up all the proteins involved in the specific process/role etc., however the font is very small and hard to be clearly read;
- Table 2 regarding cellular localization of proteins in vitro and in vivo is mistakenly named as Table 1 (instead of 2) in the legend;
- Besides tables, Figures are needed in order to complete the review and help the reader in visualizing the processes and approach to the tables more easily. An introductory figure describing the nucleocytoplasmic transport is needed and/or multiple figures to clearly show importins, exportins, channels, scaffolds etc in the pathological vs normal one;
- Authors extend on the relationship between the nucleocytoplasmic transport and the aged brain. It would be important to relate the nucleocytoplasmic transport also to other risk factors of ALS such as: neuroinflammation, metabolic diseases, dietary factors...
Author Response
Tables help to sum up all the proteins involved in the specific process/role etc., however the font is very small and hard to be clearly read
Answer: We agree that the font size is relatively small. We have enlarged the Tables as much as possible and will make sure that the size is large enough in the published version of our review.
Table 2 regarding cellular localization of proteins in vitro and in vivois mistakenly named as Table 1 (instead of 2) in the legend;
Answer: We apologize for this mistake.
Besides tables, Figures are needed in order to complete the review and help the reader in visualizing the processes and approach to the tables more easily. An introductory figure describing the nucleocytoplasmic transport is needed and/or multiple figures to clearly show importins, exportins, channels, scaffolds etc in the pathological vs normal one;
Answer: We have included two extra figures illustrating the nuclear pore complex as well as the most important nucleocytoplasmic transport processes. Given the fact that not all data are pointing into the same direction (as becomes clear from the information in the different Tables), it is at present very difficult to illustrate the pathological defects related to nucleocytoplasmic transport associated with ALS.
Authors extend on the relationship between the nucleocytoplasmic transport and the aged brain. It would be important to relate the nucleocytoplasmic transport also to other risk factors of ALS such as: neuroinflammation, metabolic diseases, dietary factors...
Answer: As there is no clear evidence about which environmental risk factors are linked to ALS, we limited our review to the effect of aging as the risk to get ALS is clearly age-related. We have inserted a sentence in the beginning of this part to make this more clear.
Round 2
Reviewer 2 Report
I appreciate the revisions made by the authors.